# A New Immortalized Human Lacrimal Gland Cell Line

**DOI:** 10.3390/cells13070622

**Published:** 2024-04-03

**Authors:** Sophie Gleixner, Ingrid Zahn, Jana Dietrich, Swati Singh, Alice Drobny, Yanni Schneider, Raphael Schwendner, Eileen Socher, Nicolas Blavet, Lars Bräuer, Antoniu-Oreste Gostian, Matthias Balk, Gundula Schulze-Tanzil, Claudia Günther, Friedrich Paulsen, Philipp Arnold

**Affiliations:** 1Institute of Functional and Clinical Anatomy, Friedrich-Alexander-Universität Erlangen-Nürnberg (FAU), 91054 Erlangen, Germany; sophie.gleixner@fau.de (S.G.);; 2Hariram Motumal Nasta & Renu Hariram Nasta Ophthalmic Plastic Surgery Services, LV Prasad Eye Institute, Hyderabad 500034, India; 3Department of Molecular Neurology, University Hospital Erlangen, Friedrich-Alexander Universität Erlangen-Nürnberg (FAU), 91054 Erlangen, Germany; 4Department of Medicine 1, University Hospital Erlangen, Friedrich-Alexander-Universität Erlangen-Nürnberg (FAU), 91054 Erlangen, Germany; 5CEITEC-Central European Institute of Technology, Masaryk University, Kamenice 5, 625 00 Brno, Czech Republic; 6Department of Otorhinolaryngology, Merciful Brothers Hospital St. Elisabeth, 94315 Straubing, Germany; 7Department of Otolaryngology, Head & Neck Surgery, Comprehensive Cancer Center Erlangen, University Hospital Erlangen, 91054 Erlangen, Germany; 8Institute of Anatomy and Cell Biology, Paracelsus Medical University, Nuremberg Prof. Ernst Nathan Str. 1, 90419 Nuremberg, Germany; 9Deutsches Zentrum Immuntherapie (DZI), University Hospital Erlangen, Friedrich-Alexander-Universität Erlangen-Nürnberg (FAU), 91054 Erlangen, Germany

**Keywords:** lacrimal gland, epithelial cells, immortalized, cell line, dry eye, tears, tear secretion

## Abstract

The lacrimal gland is crucial for maintaining ocular health by producing the aqueous component of the tear film, which hydrates and nourishes the ocular surface. Decreased production of this component results in dry eye disease, a condition affecting over 250 million people worldwide. However, the scarcity of primary human material for studying its underlying mechanisms and the absence of a cell model for human lacrimal gland epithelial cells present significant challenges. Here, we describe the generation of immortalized human lacrimal gland cell lines through the introduction of an SV40 antigen. We successfully isolated and characterized three cell clones from a female lacrimal gland donor, confirming their epithelial identity through genomic and protein analyses, including PCR, RNAseq, immunofluorescence and cultivation in a 3D spheroid model. Our findings represent a significant advancement, providing improved accessibility to investigate the molecular pathogenesis mechanisms of dry eye disease and potential therapeutic interventions. We identified the expression of typical epithelial cell marker genes and demonstrated the cells’ capability to form 2D cell sheets and 3D spheroids. This establishment of immortalized human lacrimal gland cells with epithelial characteristics holds promise for future comprehensive studies, contributing to a deeper understanding of dry eye disease and its cellular mechanisms.

## 1. Introduction

The lacrimal gland plays a crucial role in maintaining the health of the ocular surface by secreting the aqueous component of the tear film. Its secreted fluid cleanses the conjunctival sac, moistens the ocular surface, nourishes the cornea, improves optical properties and plays an important role in immune defense. Dysfunction of the lacrimal gland with decreased tear production leads to hyposecretory dry eye, one of the two major subtypes of dry eye disease [1]. Dry eye disease affects more than 250 million people worldwide, and patients suffer from symptoms such as burning, squeezing, fatigability, foreign body sensation, scratching, insensitivity, gritty sensation, visual disturbances, tearing, dryness and hypersensitivity. Objective symptoms include chronic inflammation of the eyelid margins, swollen eyelids, redness and mucus secretion. In severe cases, blindness may result from corneal opacity [2,3]. In the lacrimal gland, specialized epithelial cells (acinar cells) produce and secrete the aqueous component of the tear film, and their dysfunction is directly related to the hyposecretory form of dry eye disease [4,5,6,7,8]. However, research on the human lacrimal gland is limited due to the lack of primary tissue and the absence of a human lacrimal gland epithelial cell line [9]. Research on the cellular and molecular mechanisms underlying lacrimal gland function and dysfunction often relies on animal studies and tissues that do not resemble the human situation in all respects [10]. Insufficient human cell models similar to the epithelial cell compartment of a lacrimal gland pose a challenge to current research. The establishment of well-characterized human lacrimal gland cell lines represents a valuable tool for basic and translational investigations, allowing, for example, the study of cellular processes or the use of these cell lines in large drug screening assays.

In this study, we report the successful generation of three immortal human lacrimal gland cell lines by stably integrating an SV40 (simian virus antigen 40) DNA sequence into the genome of primary human lacrimal gland cells derived from a female patient. We introduce three cell lines derived from the same donor selected as single clones from a lacrimal gland-derived cell pool. All three cell lines express typical marker genes for epithelial cells and the lacrimal gland and form a dense epithelial cell sheet. By comparing the total RNA expression profile of the three cell lines with that of total lacrimal gland tissue, we were able to identify a notable enrichment in epithelial and myoepithelial markers. We verified the expression of typical marker genes reported before [11] for all three cell lines and showed the expression of the two important transcription factors Paired Box Gene 6 (*PAX6*) and Forkhead Box protein C1 (*FOXC1*) at the RNA and protein levels. In addition, we evaluated the growth characteristics and functional properties of these cells in both two-dimensional (2D) and three-dimensional (3D) culture models. In 3D cell culture, secretory vesicles were detected by transmission electron microscopy. We show the epithelial character of the cell lines and anticipate their great potential for future research projects. Our cells will close an important gap in lacrimal gland research and provide researchers with a valuable resource for investigating lacrimal gland biology, disease mechanisms, and therapeutic strategies. The availability of a cell line that mimics the native lacrimal gland epithelium will undoubtedly contribute to advancements in the field of lacrimal gland research and facilitate the development of novel therapeutic treatment options for dry eye disease patients.

## 2. Materials and Methods

### 2.1. Surgical Removal and Patient Description Germany

The study was approved by the institutional review board (application number: 21-374-B). After receiving signed consent from a female patient, primary human lacrimal gland tissue was obtained by complete resection of the gland at a tertiary referral medical center (Department of Otorhinolaryngology and Head and Neck Surgery, University Hospital Erlangen, 91054 Erlangen, Germany). Complete resection was performed as part of an *exenteratio orbitae* due to tumor disease. The prerequisite was that the tumor infiltrating the orbit had not exceeded the inferior and/or medial half of the orbit. This ensured a maximum distance between the tumor and the lacrimal gland. The harvested fresh tissue was immediately stored in phosphate-buffered saline (PBS) buffer for subsequent use.

### 2.2. Surgical Removal and RNA Sequencing from Lacrimal Glands

For the purpose of RNA sequencing of healthy human lacrimal glands, lacrimal gland biopsies were obtained from patients undergoing lacrimal gland debulking surgery for refractory epiphora (*n* = 4; mean age, 28.5 ± 17.8 years) [12]. All four patients were males. The project was approved by the institutional ethics committee (LCE-BHR-R-12-22-971). A trained oculoplastic specialist (S.S.) performed the tissue retrieval from the orbital lobe of the lacrimal gland as per the steps described earlier [12]. Lacrimal gland biopsies measuring 5 × 5 mm were immediately stored in an ice box after the blood was washed off and transferred to the laboratory for RNA extraction and sequencing. A modified NEBNext RNA Ultra II directional protocol was used to prepare the libraries for mRNA sequencing. The first step was selectively purifying the poly-A-containing mRNA molecules using oligo-dT-attached magnetic beads. Following purification, the mRNA was fragmented using divalent cations under elevated temperature. Next, cDNA was synthesized using reverse transcriptase and random hexamers in a first-strand synthesis reaction. Subsequently, the cDNA was converted to double-stranded cDNA, and uracil was added instead of thymine. The strand specificity was preserved by a USER enzyme-based digestion of the second strand, thereby leaving one functional strand that mapped to the DNA strand from which it was transcribed. The USER-digested single strand molecules were enriched and indexed in a limited-cycle polymerase chain reaction (PCR) followed by AMPure bead purification to create the final cDNA library for sequencing.

Prepared libraries were sequenced on Illumina HiSeqX/Novaseq at Core Facility Genomics (CEITEC Brno, Brno, Czech Republic) to generate 2 × 150 bp reads/sample. Sequenced data were processed to generate FASTQ files and were analyzed by a bioinformatician.

### 2.3. Electroporation

Cells were dissociated using Accutase^TM^ (StemPro^TM^, #A1110501, Thermo Fisher Scientific, Waltham, MA, USA) and subjected to electroporation using the nucleofector 2B (Lonza, Basel, Switzerland) with program T-023, along with the Human Stem Cell Nucleofector™ Kit 2 (Lonza). Briefly, cells were incubated with 22.2 µL of supplement 1 and 77.2 µL of supplement 2, along with 4 µg of SV40 plasmid (SV40 1: pBSSVD2005 was a gift from David Ron (Addgene plasmid # 21826; http://n2t.net/addgene:21826 (accessed on 28 March 2023; RRID:Addgene_21826)). Following electroporation, the cells were transferred to a T25 flask and cultured for further analysis.

### 2.4. Tissue Preparation and Cell Culture

The human lacrimal gland was obtained from the Otorhinolaryngology and Head and Neck Surgery department of the University Hospital Erlangen during removal of the orbit due to malignant disease. The gland tissue was freed from connective tissue and fat and washed with PBS. A small piece of the lacrimal gland was fixed with 4% paraformaldehyde (PFA) for histological analysis to ensure that the structure of the lacrimal gland was not pathological. The gland tissue was minced with a scalpel and then transferred to a culture flask containing growth-arrested 3T3 fibroblasts as a feeder layer. The explants were allowed to attach to the feeder layer before epithelial cell medium (ECM) containing Dulbecco’s Modified Eagle’s Medium/Nutrient Mixture F-12 Ham (DNEM-F12, #D8062, Merck, Darmstadt, Germany) supplemented with 10% fetal calf serum (FCS, S 0615, Bio&SELL, Feucht, Germany), 1% penicillin/streptomycin/amphotericin B (#A5955, Merck), 0.1% hydrocortisone (#H0888, Merck), 0.1% cholera toxin (#C8052; Merck), 1% sodium bicarbonate solution (25080060, Thermo Fisher Scientific), 1% adenine (#A2786, Merck), 1% T3/transferrin solution (#T8158, Merck), 0.1% insulin solution (#I5500, Merck) and 0.0002% mouse epidermal growth factor (EGF, PMG8044, Gibco, Waltham, MA, USA) was added. After 10 days, the fibroblasts were first detached by using 0.5% trypsin–EDTA solution, because the fibroblasts are less adhesive than the primary cells. The lacrimal gland cell pool was detached with 5% trypsin–EDTA solution mechanically for the first time and used for immortalization with SV40. After electroporation, cells were seeded in 96-well plates in low concentrations in order to obtain a single cell in each well. After single clone selection and an additional growth period, the clones were seeded on collagen-coated dishes. When subconfluent after 3–4 days, the cells were detached using 35 µL/cm^2^ Accutase^TM^ (Thermo Fisher Scientific, Waltham, MA, USA), seeded at 5 × 10^3^ cells/cm^2^ on collagen-coated dishes and cultured with ECM without EGF. The procedure was repeated until passages 55 (clone 1), 48 (clone 3) and 51 (clone 6). The cell clones did not undergo major changes until they reached these passages.

### 2.5. Collagen Coating

The collagen coating was performed following the manufacturer’s manual. In short, collagen coating solution (125-50; Merck, Darmstadt, Germany) or rat tail collagen coating solution (122-20; Merck) was added to the cell culture dishes at 1 mL per 10 cm^2^ surface area and incubated for 45 or 120 min at 37 °C. The solution was then aspirated, and the dishes were stored at 4 °C for up to 2 weeks. Dishes coated with rat tail collagen were washed with PBS twice before usage.

### 2.6. RNA Purification and Sequencing

After the cells were detached by mechanical and enzymatic lysis, 1 × 10^6^ cells were centrifuged at 500× *g* for 5 min, washed with PBS to remove cell debris and centrifuged again. The dry pellet was stored at −20 °C. RNA was isolated using the RNeasy Mini Kit following the manufacturer’s specifications for animal cells. Briefly, RLT buffer was added to disrupt the cells, and then 70% ethanol (EtOH) was added to allow the selective binding of RNA to the membrane. The sample was applied to the RNeasy Mini spin column and centrifuged. After washing contaminants away using RW1 and RPE buffer and centrifugation, RNAse-free water was added to elute the RNA. RNA quantification was performed using a spectrophotometer (NanoDrop 2000c, Thermo Fisher Scientific, Waltham, MA, USA). The RNA was stored at −80 °C. The RNA for RNA sequencing was purified using the RNAse-Free DNase Set following the manufacturer’s protocol for RNA isolation with DNAse digestion. RNA sequencing (*n* = 4 per clone) was performed as described below.

### 2.7. RNA Sequencing and Data Analysis

High-throughput RNA-Seq data were prepared using a Lexogen Quantseq FWD kit for Illumina with polyA selection and sequenced on an Illumina NovaSeq sequencer (run length 1 × 75 nt) at Core Facility Genomics (CEITEC Brno, Brno, Czech Republic). Bcl files were converted to fastq format using bcl2fastq v. 2.20.0.422 Illumina software for base calling. A quality check of raw single-end fastq reads was carried out by FastQC [13]. The adapter and quality trimming of raw fastq reads was performed using Trimmomatic v0.39 [14] with settings CROP:250 LEADING:3 TRAILING:3 SLIDINGWINDOW:4:5 MINLEN:35. Trimmed RNA-Seq reads were mapped against the mouse genome (mm38) and Ensembl GRCm38 v.93 annotation using STAR v2.7.3a [15] as a splice-aware short read aligner and default parameters except --outFilterMismatchNoverLmax 0.66 and --twopassMode Basic. Quality control after alignment concerning the number and percentage of uniquely and multimapped reads, rRNA contamination, mapped regions, read coverage distribution, strand specificity, gene biotypes and PCR duplication was performed using several tools, namely, RSeQC v4.0.0 [16], Picard toolkit v2.25.6 [17], and Qualimap v.2.2.2 [18].

The differential gene expression analysis was calculated based on the gene counts produced using featureCounts from the Subread package v2.0 [19] and further analyzed by the Bioconductor package DESeq2 v1.34.0 [20]. Data generated by DESeq2 with independent filtering were selected for the differential gene expression analysis due to their conservative features and to avoid potential false-positive results. Genes were considered differentially expressed based on a cutoff of adjusted *p* value ≤ 0.05 and log2(fold-change) ≥ 1 or ≤−1. Clustered heatmaps were generated from a selection of the top differentially regulated genes using the R package pheatmap v1.0.12 [21], volcano plots were produced using the ggplot2 v3.3.5 package [22], and MA plots were generated using the ggpubr v0.4.0 package [23].

### 2.8. cDNA Synthesis

First-strand cDNA synthesis was performed using RevertAid H Minus Reverse Transcriptase (200 U/µL; EP0451; Thermo Fisher Scientific) following the manufacturer’s manual. Briefly, DEPC-treated water was added to 4 µg of RNA to reach a final volume of 24 mL. Two microliters of oligo(dT) primer was added and heated at 65 °C for 5 min. After the addition of 14 µL of mastermix containing 8 µL of 5× reaction buffer, 4 µL of dNTP Mix, 1 µL of RNAse inhibitor and 1 µL of ReverseAid H Minus Reverse RT, the samples were heated at 42 °C for 1 h, followed by 10 min at 70 °C to inactivate the reverse transcriptase. cDNA was stored at −20 °C.

### 2.9. Reverse Transcription (RT)-PCR and Gel Electrophoresis

Primers and corresponding annealing temperatures utilized for PCR are indicated in Table 1. PCR was performed using Taq DNA Polymerase (5 U/µL, native, without BSA; EP0282; Thermo Fisher Scientific) following the manufacturer’s manual. In short, 18 µL of master mix containing 2 µL of 10× Taq buffer, 2 µL of 50 mM MgCl_2_, 1 µL of dNTP mix, 0.5 µL of forward and reverse primers, 11.8 µL of nuclease-free water and 0.2 µL of Taq DNA polymerase was added to 2 µL of cDNA. All reactions were performed in the thermal cycling program consisting of an activation step (95 °C, 5 min) followed by 39 cycles of denaturing (95 °C, 30 s), annealing (59–63 °C, 30 s) and extension (72 °C, 30 s). PCR products were identified after agarose gel electrophoresis (500 A/90 V) on a 1.5% or 2% agarose gel and staining with GelRed^®^ (41003; Biotium, Fremont, CA, USA).

### 2.10. Semiquantitative RT-PCR Analysis

The images of the agarose gels were captured under ultraviolet (UV) light with the gel documentation system Dark Hood DH-50 (Biostep, Jahnsdorf, Germany) and then analyzed using the image analysis software ImageJ, version 1.50i. The bands were selected by drawing rectangles covering the specific bands of interest and then plotted. A line was drawn to connect the maximum values (gel) around the minimum value (band) of the curve to measure the density of the bands. Then, the inside of the curves was selected with the wand tool, giving the area under the curve in pixels. The values for each gene and clone were normalized to the average expression on each gel. The values were then visualized using GraphPad Prism version 10.0.0 for Windows, (GraphPad Software, San Diego, CA, USA, www.graphpad.com). The reagents, size and thickness of the agarose gels and other conditions were kept constant.

### 2.11. Fluorescence Immunostaining of PAX6,FOXC1/2, CLDN5, KRT19 and OCLN

Cells were seeded at 5 × 10^3^ cell/cm^2^ on collagen-coated glass cover slips and cultivated for 3 days. Cells were fixed with 4% paraformaldehyde (PFA; 0335.3; Carl Roth, Karlsruhe, Germany) for 15 min at room temperature (RT) followed by three washing steps with PBS. The cells were blocked in PBS and Tween-20 with 1% (*w*/*v*) nonfat dry milk for 1 h at RT followed by incubation with 1:50 PAX6 (AF8150; R&D Systems, Minneapolis, MN, USA), 1:500 FOXC1/2 (CPA3482; Cohesion, Suzhou, China), CLDN5 (MA532614, Invitrogen, Waltham, MA, USA), 1:50 KRT19 (M0888; Dako, Santa Clara, CA, USA) or OCLN (33-1500, Thermo Fisher Scientific, Waltham, MA, USA) primary antibody dilutions overnight at 4 °C in a humidified chamber. After 3 washes with Tris-buffered saline (TBS: 0.05 M Tris, 0.015 M NaCl, pH 7.6) and Tween-20, the cells were incubated with 1:200 donkey anti-sheep (A-11015, Molecular Probes Inc., Eugene, OR, USA), 1:1000 donkey anti-rabbit (A-21206; Invitrogen, Waltham, MA, USA) or goat anti-mouse (A-11029; Invitrogen) secondary antibodies for 1 h at RT and washed 3 times. The nucleus was counterstained with 4′,6-diamidino-2′-phenylindol (DAPI; D9664; Merck, Darmstadt, Germany) at a 1:1000 dilution for 10 min at RT, and the cells were washed. Finally, the coverslips were mounted with fluorescence mounting medium (S3023; Dako, Santa Clara, CA, USA). Images were acquired using an inverted fluorescence phase contrast microscope (Keyence BZ-X810, Keyence, Neu-Isenburg, Germany). The utilized antibodies are indicated in Table 2.

### 2.12. Fluorescence Immunostaining of SV40

The cells were cultured in the same way as for the other immunofluorescence stains. Then, the cells were washed three times with TBS before being incubated with protease-free donkey serum blocking and permeabilization buffer (5% in TBS with 0.1% Triton X-100) for 20 min at RT. Subsequently, cover slips were incubated with the primary antibody (SV40 T antigen mouse anti-human, Merck-Millipore, Darmstadt, Germany, 1:50) overnight at 4 °C in a humidifier chamber. Samples were rinsed three times with TBS prior to incubation with donkey anti-mouse cyanine (Cy)3 secondary antibody (Invitrogen, Waltham, MA, USA) combined with Alexa Fluor 488 phalloidin (Santa Cruz Biotechnologies, Santa Cruz, CA, USA) (diluted 1:200 and 1:100 in TBS with 0.1% Triton ×100 and 5% donkey serum) to visualize cytoskeletal F-actin organization for 1 h at RT in a humidifier chamber. During this incubation step, cell nuclei were counterstained using 4′,6-diamidino-2′-phenylindol (DAPI, Roche, Mannheim, Germany). Labelled cells were washed three times with TBS before being covered with Fluoromount mounting medium (Southern Biotech, Biozol Diagnostica, Eching, Germany). Photos were taken using confocal laser scanning microscopy (TCS SPEII; Leica, Wetzlar, Germany). The utilized antibodies are indicated in Table 2.

### 2.13. Fluorescein Isothiocyanate (FITC)-Dextran Permeability Assay

An equal number of cells (9 × 10^4^ per cm^2^) were seeded on collagen-coated Transwell inserts (0.4 µm of polyethylenterephthalate membrane, translucent, 24-well plate; cellQART). The cells were cultivated in ECM for 2 or 4 days. To visualize the permeability of the cell layer, FITC–dextran (53379; Merck, Darmstadt, Germany) was added to the upper chamber (1 mg/mL). After 30 min, the Transwells were removed, and fluorescence in the medium of the bottom chamber was analyzed with a microplate reader (CLARIOstar, BMG LABTECH GmbH, Ortenberg, Germany).

### 2.14. Transepithelial Electrical Resistance (TEER) Measurement

An equal number of cells (9 × 10^4^ per cm^2^) were seeded on Transwell inserts (0.4 µm polyethylenterephthalate membrane, translucent, 24-well plate; cellQART) with 300 µL of ECM. One milliliter of ECM was added to the lower chamber. Prior to measurement, the chopstick electrode (MERSSTX01; Merck-Millipore, Darmstadt, Germany) was sterilized in 70% EtOH for 10 min, then rinsed with PBS. Two consecutive resistance measurements were taken in each well at RT. The blank value was measured in six wells of Transwell inserts without cells. TEER was measured after 3, 5, 7 and 10 days using a voltohmmeter (Millicell-ERS2; Merck-Millipore, Darmstadt, Germany). The blank value was averaged and subtracted. The resulting value was then multiplied by the surface area of the well (0.3 cm^2^) to obtain the resistance values in [Ω × cm^2^].

### 2.15. Spheroid Culture

A total of 2 × 10^4^ cells in 200 µL were seeded in ultralow attachment 96-well round bottom plates (7007; Corning, Corning, NY, USA). The cells were incubated (37 °C, 5% CO_2_, 95% humidity) and showed spheroid formation after 24 h, which was visually confirmed. The spheroids were harvested after 8 days. Spheroids for PCR analyses and immunofluorescence staining were further processed similarly to the cells in the 2D growth culture.

### 2.16. Histological Staining

Spheroids were fixed using Histogel™ (HG-4000-01; Epredia™, Kalamazoo, MI, USA) according to the manufacturer’s instructions. The fixed cells were embedded in paraffin, sectioned and placed on microscope slides. For deparaffinization, slides were first incubated in xylene solution for 20 min and then in EtOH (100/100/96/80/70%) for 3 min per concentration.

For hematoxylin–eosin staining, after washing with distilled water, the slides were incubated in hemalum for 10 min, washed in tap water for 15 min, and washed with distilled water. The slides were then incubated in eosin for 4 min. The staining was stopped with distilled water. Slides were then incubated in ascending concentrations of EtOH (70/80/96/96/100/100/100%), followed by incubation in xylene solution for 20 min. Slides were coverslipped with Entellan.

For Heidenhain azan trichrome staining, after washing with distilled water, the slides were incubated for 15 min in heated azocarmine (60 °C for 1 h, 0.1 g of azocarmine with 100 mL of distilled water). The slides were washed with distilled water and then incubated in an aniline blue-orange G solution (0.5 g of aniline blue, 2.0 g of orange G, 300 mL of distilled water, 8 mL of acetic EtOH) for 15 min. The slides were then washed with ethanol (96%) and incubated in EtOH (100%) and xylene solution for 3 min each. The slides were coverslipped with Entellan.

### 2.17. Spheroid Analysis Area, Perimeter, Circularity, and Roundness

To determine the area of the spheroids, microscopic images of the spheroids were converted to black-and-white images to represent each spheroid as a black object on a white background. Thereafter, these images were analyzed using two in-house scripts for measuring the size of objects in an image with OpenCV. A script measures the number of black pixels, which are synonymous with the size of the spheroids. In addition, the number of pixels building the edge of the spheroids is measured, giving the perimeter. The other script draws a rectangle with the minimum area around the black object and calculates the edge lengths of the rectangle (Appendix A). The circularity, indicating the similarity of the object to a circle, was calculated with the following formula: $circularity=4\pi \left(\frac{\left[area\right]}{{\left[perimeter\right]}^{2}}\right)$. For exactly circular objects, both edges of the rectangle have the same length, leading to a circularity of ~1, while for elliptical objects, the edge lengths would be different, leading to a lower circularity. Roundness, indicating the smoothness of the object, was calculated with the following formula: $roundness=4(\frac{\left[area\right]}{\pi \times {\left[major\;axis\right]}^{2}})$.

### 2.18. Three-Dimensional Culture in Extracellular Matrix Environment

A total of 3–8 spheroids grown for 8 days were seeded in 25 µL of 1:2 diluted Cultrex^®^ RGF BME, Type 2 and basal culture medium (BCM) in a 48-well plate. BCM contains Advanced DMEM/F12 (12634028, Invitrogen), HEPES (10 mM, PAA), GlutaMax (2 mM, Invitrogen, Waltham, MA, USA), penicillin (100 U/mL, Gibco, Waltham, MA, USA), and streptomycin (100 μg/mL, Gibco, Waltham, MA, USA). Spheroids were incubated at 37 °C for 1 h to allow the diluted extracellular matrix to harden. An amount of 250 µL of ECM or ECM with 50 ng/mL of EGF and 100 ng/mL of Fibroblast Growth Factor 10 (FGF10) proteins was added. For 96 h, images were taken every two hours.

### 2.19. Transmission Electron Microscopy

8-day grown spheroids were fixed in glutaraldehyde-containing buffer and were thereafter treated as small tissue samples. Further processing was performed as described previously [24]. Images were acquired on a JEOL1400Plus (JEOL Germany, Freising, Germany) operating at 120 kV acceleration voltage.

## 3. Results

### 3.1. Generation of Immortalized Lacrimal Gland Cell Lines

Primary human lacrimal gland tissue was obtained after receiving signed consent from an adult female patient by surgical resection at a tertiary referral medical center (Department of Otorhinolaryngology and Head and Neck Surgery, Friedrich-Alexander-Universität Erlangen-Nürnberg (FAU), Erlangen, Germany) and was approved by the institutional review board (application number: 21-374-B). Seeding the minced tissue on a murine fibroblast feeder layer resulted in primary cell outgrowth. After ten days, the murine feeder layer was removed with a trypsin concentration that would not detach the outgrowing more adhesive lacrimal gland cells. In the second step, these cells were then detached with a higher concentrated trypsin solution and harvested as a pool, containing all cell types of the lacrimal gland. A SV40-carrying plasmid was then introduced into the cells by electroporation. After an additional proliferation period, we selected individual cell clones from the pool for further characterization (Figure 1A). We performed reverse transcription polymerase chain reaction (RT-PCR) for different marker genes to select clones that showed the expression of epithelial cells. For comparative reasons, we included a human corneal epithelial cell line (CEC), human mesenchymal stem cells (hMSC) and a fibroblast line (3T3 fibro) (Figure 1B). We included markers for epithelial, mesenchymal and endothelial cells and fibroblasts (Figure 1C) and identified clones 1, 3 and 6 to express E-cadherin (*CADH1*), sialomucine (*CD34*), smooth muscle actin (*ACTA2*), vimentin (*VIM*) and the two transcription factors Forkhead Box C1 (*FOXC1*) and Paired Box Gene 6 (*PAX6*), which are important for the differentiation of cells in various types of ocular epithelium (Figure 1B,C). Compared to clones 1, 3 and 6, there was a marked difference in gene expression for clone 5, which showed similarity to the expression pattern of the fibroblast control line (strong for *ACTA2* and *VIM*). Thus, we decided to utilize clone 5 as a fibroblast control for the next experiments. Then, we tested different clones for the expression of the inserted *SV40* gene by RT-PCR (Figure 1D). Clones 1, 3 and 6, the lacrimal gland cell pool and the CEC line were positive for *SV40*, whereas 3T3 fibroblasts, hMSC, clone 5 fibroblasts and the H_2_O control showed no signal for *SV40* (Figure 1D). To demonstrate the localization of the SV40 gene product in the nucleus, we stained clones 1, 3 and 6 for SV40 and included clone 5 as a negative control. To visualize the cell structure, we included F-actin (Figure 1E). Indeed, we identified the SV40 gene product in the nucleus of clones 1, 3 and 6 but not in the nucleus of clone 5. The formation of a dense two-dimensional cell sheet is one hallmark of epithelial cells, and we tested clones 1, 3 and 6 for this property. Moreover, we used scanning electron microscopy to show sheet formation after the cells reached confluence (Appendix A). All three cell lines formed two-dimensional cell sheets and did not grow as stacks or islands (Appendix A). To test the density and tightness of the cell sheet, we cultured cells on membranes in cell culture inserts (Appendix A). After two and four days, we added a fluorescein isothiocyanate (FITC)–dextran-containing solution to the upper chamber and allowed diffusion to the lower chamber. After 30 min, we took samples from the lower chamber and measured the FITC–dextran signal in a plate reader. All three cell clones, clones 1, 3 and 6, formed tight barriers over the course of four days, and the amount of FITC–dextran was only approximately 10% of the amount of an empty cell culture insert (Appendix A). As an additional analysis of the tightness of the cell sheet, we measured the transepithelial electrical resistance (TEER) after three, five, seven and ten days (Appendix A). The values for all three cell clones showed a slight increase, ranging between 20 and 30 Ω × cm^2^ (Appendix A). Thus, we decided to include all three cell clones in our further analysis and performed total RNA sequencing.

### 3.2. Total RNAseq of Immortalized Human Lacrimal Gland Cells

To obtain a full picture of gene expression for the three SV40-positive cell clones, we performed total RNA sequencing (RNAseq) for four replicates of each cell clone and compared them to human lacrimal gland tissue from four donors. We then performed a differential expression analysis and found similar genes within the top 20 up- or downregulated genes in all three cell clones (Figure 2A). We identified that metallothionein 2A (*MT2A*), DNA topoisomerase II alpha (*TOP2A*) and two different genes for structural maintenance of chromosomes (SMCs), *SMC2* and *SMC4*, were upregulated in all three cell clones, whereas four ribosomal protein (RPL) genes (*RPL3, 7, 10, 21*), eukaryotic translation elongation factor 1 alpha 1 (*EEF1A1*) and SRP receptor subunit alpha (*SRPRA*) were downregulated. To identify regulated pathways, we analyzed the top 100 up- and downregulated genes for each clone using HumanMine’s (humanmine.org) GOterm gene analysis tool. We identified similar pathways significantly upregulated in all three cell clones, mostly related to cell proliferation, cell division and chromosome segregation (Figure 2B). Among the downregulated GOterms, we identified mainly those associated with cellular, structural or tissue homeostasis (Figure 2B). Next, we compared the expression levels of marker genes for epithelial cells, endothelial cells, hMSC, myoepithelial cells and fibroblasts between the cell clones and total lacrimal gland tissue (Figure 2C; a list of gene names can be found in Appendix A). We identified enriched expression of typical epithelial and myoepithelial genes but not endothelial, mesenchymal stem or fibroblast markers. As the enrichment of epithelial and myoepithelial cells showed high similarity among all three cell clones tested, we decided to continue with all three and assessed the expression of marker genes of lacrimal gland epithelial cells as reported from primary tissue [11].

### 3.3. Validation of Lacrimal Gland Markers at the Transcriptional and Protein Levels

First, we analyzed the expression of all genes identified as markers for lacrimal gland epithelial cells by the group of Hans Clevers [11] in all three clones from the RNAseq datasets (Figure 3A, a list of gene names can be found in Appendix A). We found that most marker genes were expressed in the individual cell clones and decided to validate the RNAseq data with a more specific/quantitative method. Thus, we performed RT-PCR in a clone-dependent manner and measured the expression of the epithelial marker genes *AQP5*, *CST6*, *CSTB*, *FOXC1*, *MYL9* and *PAX6* and the myoepithelial marker *ACTA2* (Figure 3B, a list of gene names can be found in Appendix A). As controls, we included a fibroblast line and the corneal epithelial cell line mentioned above. When we analyzed multiple experiments semiquantitatively, we identified similar expression between our three cell clones and the corneal epithelial cells but large differences when compared to the fibroblast line (Figure 3C). Almost all epithelial cell markers were significantly increased in the three cell clones when compared to the fibroblast line, and for the myoepithelial marker *ACTA2*, we measured a significant increase in fibroblasts when compared to the epithelial cell lines (Figure 3C). The reference gene actin beta (*ACTB*) showed no differences among cell lines (Figure 3C). We also tested the expression of genes that encode important secreted proteins such as lactotransferrin (*LTF*) and the polymeric immunoglobulin receptor (*PIGR*). *LTF* is an antimicrobial peptide, and *PIGR* polymerizes IgA that lacrimal gland epithelial cells take up at the basolateral side and then transcytoses them for secretion at the apical side [25]. The expression of these two genes seemed to be dependent on the cell passage (Appendix A). Thus, we also measured the expression of these genes and lipocalin 2 *(LCN2*), at different levels of cell confluence (50% and 100%) to assess whether the formation of a 2D cell sheet (at 100% confluence) might induce the expression of epithelial markers (Appendix A). Although not significant, a clear trend could be seen for clones 3 and 6 for the expression of *LTF*, where higher confluence induced gene expression (Appendix A). For *LCN2*, this effect was observed in all clones (Appendix A). All values were normalized to actin beta (*ACTB*). To demonstrate the localization of the transcription factors FOXC1 and PAX6 in the nucleus, we stained clones 1, 3 and 6 for both proteins. The expression of FOXC1 and PAX6 in the nucleus was visualized for all three clones by immunofluorescence staining (Figure 3D). Additionally, we stained for epithelial markers occludin (OCLN), claudin (CLDN) and keratin 19 (KRT19) and found expression for these in different cell clones (Appendix A).

### 3.4. Utilization of Immortalized Cells in a 3D Spheroid Model

Next, we applied a 3D spheroid cell model to the three lacrimal gland cell clones. One advantage is cell growth without any surface coating, making the results more reproducible. For this, we seeded 2 × 10^5^ cells on ultralow attachment plates and documented their growth behavior for up to 29 days (Appendix A). After 24 h, the cells began to form spheroids (Figure 4A). These spheroids became smaller, rounder and denser over the course of 29 days (Figure 4A and Appendix A). To compare the different clones and to measure parameters such as area, perimeter, circularity and roundness in an unbiased approach, we applied two scripts developed in-house (see the Materials and Methods Section 2.17, Appendix A). It became clear that all three clones showed a similar behavior that was marked by an initial densification of the spheroids over the first 5–7 days that was indicated by a reduction in spheroid area and perimeter (Figure 4B). The circularity of the samples increased during the densification period, underlining the globular organization of spheroids (Figure 4B). The roundness of spheroids remained similar during the entire observation time (Appendix A). To analyze the spheroids in more detail, we fixed and embedded them in a gel matrix for histologic sectioning. The fluorescence staining of nuclear DNA using DAPI showed a homogenous distribution of nuclei (Figure 4C). Hematoxylin–eosin (HE) and azan staining revealed the arrangement of the cells in the 3D spheroid model, and it appears that cell density was higher in the outer layers of the spheroids (Figure 4C). Azan staining showed no prominent deposition of the extracellular matrix that would stain blue. Next, we assessed the expression of a subset of epithelial marker genes (Figure 4D). For all three cell clones in the 3D spheroid model, the expression of *CSTB*, *FOXC1*, *LTF*, *MYL9*, *PAX6* and *PIGR* was confirmed, whereas little or no expression was detected for *AQP5*, *CST6* and *LYZ* (Figure 4E). Magnified images of the spheroids showed a stretched morphology of the border cells and, thus, we conclude an epithelial-like cell growth for them (Figure 4F). Additionally, we seeded spheroids in an extracellular matrix. After 24 h, clone 3, especially, showed budding, with constrictions that deepened and formed larger buds after 48 h, as expected for epithelial cells in spheroid models (Appendix A). Budding was observed regardless of whether the medium was with or without growth factors. The independence from growth factors may be due to the continuous proliferation caused by the SV40 insertion. Spheroids grown in the medium with growth factors showed more cells evading from the spheroid, which might represent mesenchymal outgrowth (Appendix A insert). Transmission electron microscopy (TEM) of the spheroids grown for 8 days showed secretory granules in all three cell clones (Figure 4G, arrowheads). In addition to secretory granules, we also identified glycogen deposits inside the cells, which were previously also described as a marker for lacrimal gland epithelial cells in alpaca [26]. As in light microscopy, the border cells of the spheroids also showed a stretched epithelial appearance in electron microscopic images (Figure 4G, star).

## 4. Discussion

We stably inserted an *SV40* gene into primary human lacrimal gland cells for immortalization, as this strategy was successfully used before for corneal epithelial cells [27]. We selected three single-cell clones after the immortalization and confirmation of epithelial marker gene expression, such as *PAX6* and *FOXC1* [11,28,29]. The detection of *SV40* in the cell nucleus [30] and its expression by RT-PCR demonstrated successful immortalization of the cells via electroporation. The three cell clones grew as cell sheets and formed a dense epithelial barrier, one of the hallmarks of epithelial cells [31]. We could show this barrier function in FITC–dextran and TEER measurement experiments. TEER values were comparable to epithelial cells from salivary glands (Gl. parotidea, submandibularis and lingualis) [32,33]. To categorize and characterize the cell types in more detail, we utilized RNAseq, specific RT-PCR experiments and immunofluorescence analyses. The RNAseq data GOterm analysis showed an upregulation of pathways responsible for cell proliferation, such as cell cycle progression, cell division and chromosome segregation. Thus, cells are more prone to cell division when compared to lacrimal gland tissue, which is most likely caused by *SV40* immortalization. At the same time, homeostasis pathways such as anatomical structure homeostasis and tissue homeostasis are downregulated when compared to the entire lacrimal gland tissue. These findings appear to be consistent across all three cell lines. We showed the expression of lacrimal gland-specific genes, as suggested by the group of Hans Clevers [11], in RNAseq data and showed specific expression in RT-PCR experiments, demonstrating a marked difference from the fibroblast control line. In more detail, selected clones showed enriched expression of the epithelial marker genes *CSTB* and *CST6* [11] and the myoepithelial markers *HAS2* (clone 2), *FN1*, and *PALLD* [34,35,36,37]. Endothelial, mesenchymal and fibroblast marker genes were downregulated in all three cell clones compared to the total lacrimal gland tissue. Thus, we suggest that we obtained epithelial cells that might also have a partial myoepithelial and mesenchymal character. However, we can show that secreted factors such as *LTF* and *PIGR* are expressed in a passage- and confluence-dependent manner, as was seen for other epithelial cells as well [38]. We also tested our cell clones in a 3D spheroid model that we applied previously [30], as 3D spheroids hold some advantages over 2D cell culture. First, no coating of the cell culture dish is necessary; second, cells show in vivo metabolic responses [39]; and third, they can be utilized in all stages of drug discovery [40]. TEM imaging of the spheroids showed secretory vesicles in the cytoplasm of inner cells. This strongly suggests secretory potential and indicates a (partial) epithelial character of the cells. After the embedding of spheroids in the extracellular matrix (matrigel), we identified the formation of buds, a typical sign for epithelial cells in 3D cell culture models [41].

Although we described the first available cell model for human lacrimal gland cells here, there are also limitations to be considered. We detected not only enriched epithelial markers but also myoepithelial and mesenchymal markers. Thus, we cannot exclude a mixed cell entity, especially at lower densities. After reaching full confluence, cells express higher levels of *LTF*, which might indicate their ability to react to contact inhibition-induced proliferation arrest and increasing differentiation [42]. We also observed some differences in gene expression analysis within the three single clones selected, although from the same patient. However, this might serve as an advantage, as only the effects observed in all three cell lines represent a common mechanism among epithelial cells from lacrimal gland tissue. It is important to note that the analysis mainly focused on gene expression, which may not necessarily reflect actual protein expression. To further enhance the characterization, proteomics technology could be employed to detect the expressed proteins. We propose that our cell lines serve as starting points for lacrimal gland researchers, whose results have to be confirmed in primary human cells. From our experiments, we would suggest that our selected clones display more epithelial features in 3D spheroid models when compared to 2D cell cultures. The main advantage of our newly established cell model is the availability of large cell quantities due to constant proliferation. In primary cells, proliferation stops after a few cell divisions, making application in larger drug screenings impossible. Future applications could also include the application of these cells in organoid models; however, proliferation must be limited after organoid generation.

## Figures and Tables

**Figure 1 cells-13-00622-f001:**
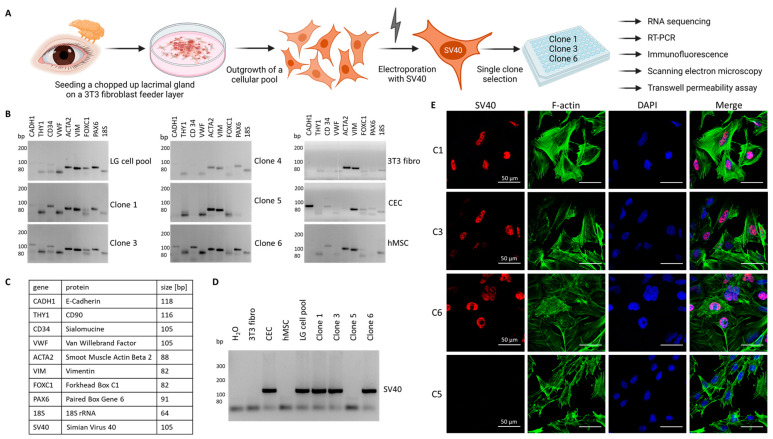
Selection of immortalized human lacrimal gland cells. (**A**) Cartoon illustrating the workflow from the lacrimal gland to the single clones. The figure was partly created with BioRender.com. (**B**) Agarose gels with the amplification products of RT-PCR against marker genes tested in the lacrimal gland cell pool (LG cell pool), single clones (clone 1, 3, 4, 5 and 6), a murine 3T3 fibroblast line (3T3 fibro), human corneal epithelial cells (CEC) and human mesenchymal stem cells (hMSC). (**C**) List of the genes tested in (**B**) with gene names, names of the respective protein product and the expected amplicon size. (**D**) Agarose gel with the amplification product of the inserted *SV40* gene. (**E**) Immunofluorescence image of clones 1 (C1), 3 (C3), 5 (C5) and 6 (C6) stained against *SV40* (red), *F-actin* (green) and the nucleus (blue).

**Figure 2 cells-13-00622-f002:**
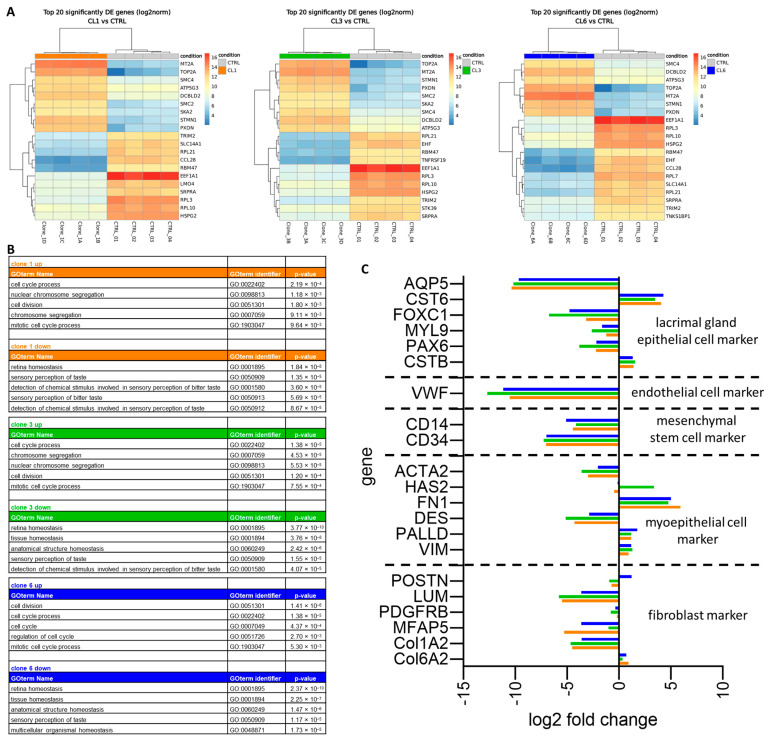
Total RNA sequencing of immortalized human lacrimal gland cells. (**A**) Differential expression analysis of the top 20 up- and downregulated genes for clone 1 (CL1), clone 3 (CL3) and clone 6 (CL6) compared to human lacrimal gland tissue of 4 donors (CTRL). (**B**) Top 5 up- and downregulated pathways using HumanMine’s (humanmine.org) GOterm gene analysis tool with GOterm identifier and *p* value. (**C**) Comparison of gene expression levels of marker genes for epithelial cells, endothelial cells, mesenchymal stem cells, myoepithelial cells and fibroblasts between clone 1 (orange), clone 3 (green), clone 6 (blue) and total lacrimal gland tissue. A list of marker genes can be found in Appendix A.

**Figure 3 cells-13-00622-f003:**
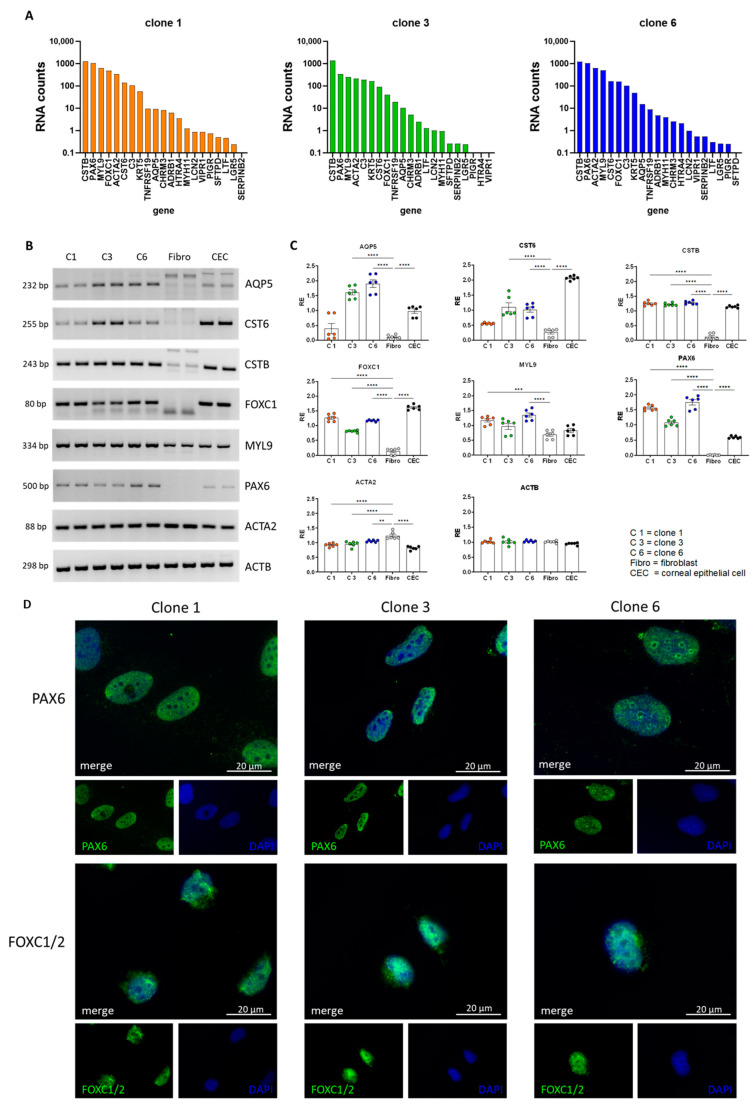
Validation of lacrimal gland markers at the transcriptional and protein levels. (**A**) Gene expression of lacrimal gland epithelial marker genes in clones 1, 3 and 6 (RNA counts imply counts per million). A list of all gene abbreviations can be found in Appendix A. (**B**) Agarose gels with the amplification products of RT-PCR against marker genes tested in clone 1 (C1), clone 3 (C3), clone 6 (C6), human fibroblasts (Fibro) and human corneal epithelial cells (CEC). A list of marker genes can be found in Appendix A. (**C**) Semiquantitative comparison of gene expression after RT-PCR and gel electrophoresis. A list of marker genes can be found in Appendix A. Data are means ± SEMs ** *p* < 0.01, *** *p* < 0.001, and **** *p* < 0.0001, two-way ANOVA, *n* = 6. (**D**) Immunofluorescence images of clones 1, 3 and 6 stained with DAPI (blue), Paired Box Gene 6 (*PAX6*, green, upper row) and Forkhead Box C1/C2 (*FOXC1/2*), green, bottom row.

**Figure 4 cells-13-00622-f004:**
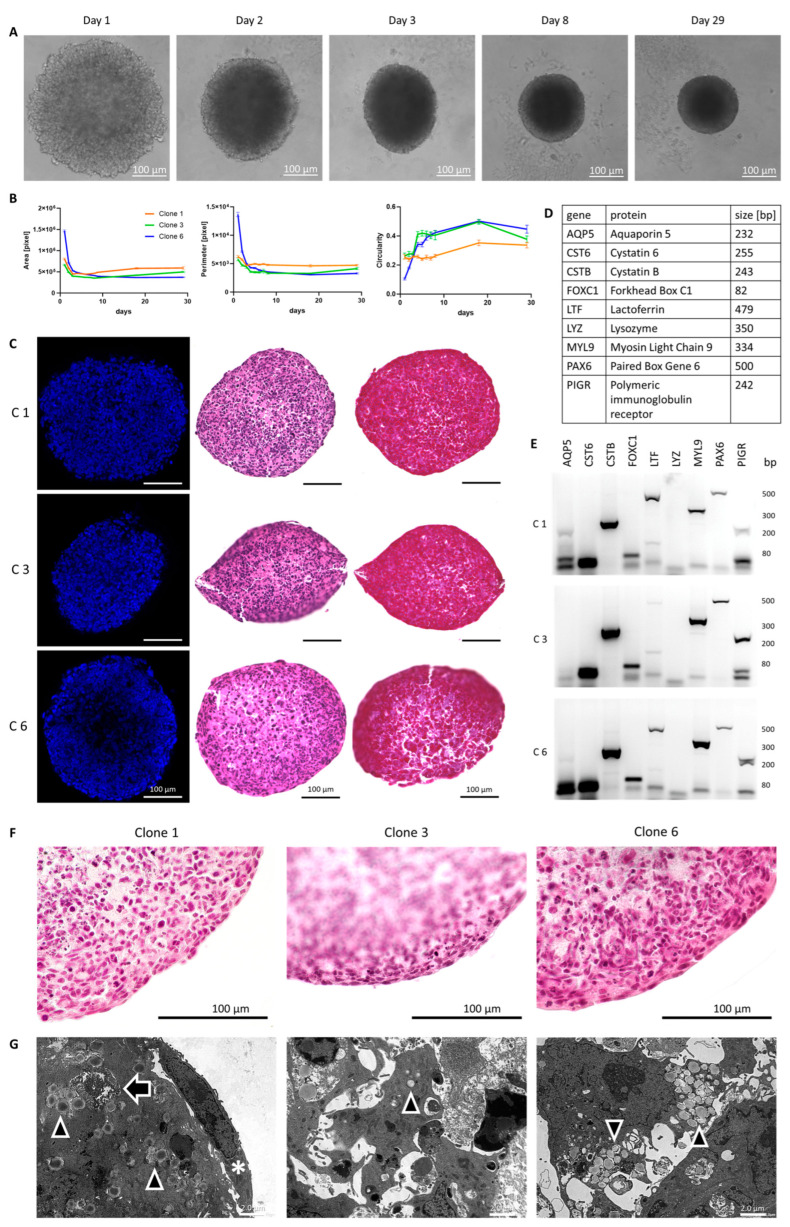
Analysis of the 3D spheroid cell culture model with immortalized lacrimal gland cells. (**A**) Transmitted light microscopy during the culture period of 3D-grown cells of clone 6. (**B**) Analysis of the area, perimeter and circularity of the spheroids of clones 1 (orange), 3 (green) and 6 (blue) over the course of a 29-day growth period. Data are means ± SEMs. [Day 1–8 *n* = 24, day 18 *n* = 9, and 29 *n* = 9]. (**C**) Image of 8-day-grown spheroids of clones 1 (C1), 3 (C3) and 6 (C6) with immunofluorescence staining against the nucleus (blue, left column), hematoxylin–eosin (middle column) and azan staining (right column). (**D**) List of the genes tested in (**E**) with gene names, names of the respective protein product and the expected amplicon size. (**E**) Agarose gels with the amplification products of RT-PCR against marker genes tested in clones 1, 3 and 6. (**F**) Magnified image of 8-day grown spheroids of clone 1, 3 and 6 with hematoxylin–eosin staining. (**G**) Transmission electron microscopy (TEM) image of 8-day grown spheroids with secretory granules (arrowheads), glycogen aggregation (arrow) and spheroid border cell (star).

**Table 1 cells-13-00622-t001:** Primers RT-PCR.

Gene	Primer Sequence	Product Size (Base Pairs)	Annealing Temperature [°C]
*ACTA2*	GCCGAGATCTCACCGACTAC TCACGGACAATCTCACGCTC	88	60
*ACTB*	GATCCTCACCGAGCGCGGCTACA GCGGATGTCCACGTCACACTTCA	298	60
*AQP5*	TCCATTGGCCTGTCTGTCAC CTTTGATGATGGCCACACGC	232	63
*CADH1*	AGGGGTTAAGCACAACAGCA ACGACGTTAGCCTCGTTCTC	118	60
*CD34*	TAGACTGTGCAGTGATGTGGT GGCAGACTTGGCTAAAGGTCC	105	60
*CD90*	TGGATTAAGGATGAGGCCCG GGGGAGGTGCAGTCTGTATT	116	60
*CST6*	GGCAGCAACAGCATCTACTACTT ACAGTTGTGCTTTAGGAGCTGAG	255	59
*CSTB*	CGTGTCATTCAAGAGCCAGG CGCTCTGGTAGACGGAGGAT	243	59
*FOXC1*	TCGGCTTGAACAACTCTCCA GACGTGCGGTACAGAGACTG	82	60
*hu18S*	GGAGCCTGAGAAACGGCTA TCGGGAGTGGGTAATTTGC	64	60
*LTF*	CAGACCGCAGACATGAAACT TTCAAGAATGGACGAAGTGT	479	60
*LYZ*	CTCTCATTGTTCTGGGGC ACGGACAACCCTCTTTGC	350	60
*MYL9*	ACCCACCAGAAGCCAAGATG GCGTTGCGAATCACATCCTC	334	63
*PAX6*	TAACCTGCCTATGCAACCCC ATAACTCCGCCCATTCACCG	91	60
*PAX6*	AGTTCTTCGCAACCTGGCTA TGAACGTGCTGCTGATAGGA	500	57
*PIGR*	AATGCTGACCTCCAAGTGCTAAAG ATCACCACACTGAATGAGCCATCC	242	60
*VIM*	GCTTCAGAGAGAGGAAGCCG AAGGTCAAGACGTGCCAGAG	82	60
*VWF*	AGAACAGATGTGTGGCCCTG CTTCCGGTCCTGACAGACAC	113	60

**Table 2 cells-13-00622-t002:** Antibodies for immunofluorescence staining.

Antibody	Host Species	Dilution	Company
PAX6	Sheep	1:50	R&D Systems, Minneapolis, MN, USA
FOXC1/2	Rabbit	1:500	Cohesion, Suzhou, China
SV40	Mouse	1:50	Merck-Millipore, Darmstadt, Germany
F-Actin	Mouse	1:200	Invitrogen, Waltham, MA, USA
CLDN5	Rabbit	1:100	Invitrogen, Waltham, MA, USA
KRT19	Mouse	1:50	DAKO, Santa Clara, CA, USA
OCLN	Mouse	1:250	Thermo Fisher Scientific, Waltham, MA, USA

## Data Availability

The RNAseq data have been deposited in the Gene Expression Omnibus and are accessible through the accession number GSE239464. All other data are available in the main text or in the Appendix A.

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
