# Peer review of "A New Immortalized Human Lacrimal Gland Cell Line"

_cells, 2024, doi:10.3390/cells13070622_

Round 1

Reviewer 1 Report

Comments and Suggestions for Authors

In this manuscript, authors describe the obtention and characterization of 3 immortalized cell clones of lacrimal epithelial cells that could become very important experimental models to study lacrimal gland functions and alterations, an issue that has scarcely studied due to the lack of adequate models to approach both the physiology and pathology of lacrimal glands. On the basis of the above, the results obtained by authors are highly relevant.

MAJOR ISSUSES:

1)    Authors should describe the characteristics of the SV40 plasmid, including the promoter used for expressing SV40LT antigen to promote the immortalization of cells.

2)    Authors wrongly remark that PAX6 and FOXC1 are lacrimal gland epithelial cell markers. This should be corrected since PAX6 is a differentiation marker of various ocular cell types; it is expressed in lens, retina, cornea, conjunctiva among others (Kodama & Eguchi, 1994, Curr Opin Genet Dev. 4(5):703-708; Koroma et al., 1997, Invest Ophthalmol Vis Sci. 38(1):108-210; Nishina et al., 1999, Br J Ophthalmol. 83(6):723-727; Funderburgh et al., 2005, FASEB J. 2005 19(10):1371-1373.). Is a similar case for FOXC1 (Wang et al, 2001, Mol Vis. 7:89-94; Ittner et al., 2005, J Biol. 4(3):11), which is also essential for the normal expression of corneal epithelial differentiation (Li et al., 2021, Signal Transduct Target Ther. 6(1):5; Yang et al., 2023, Exp Eye Res. 234:109599).

3)    To demonstrate that the immortalized cells belong to an epithelial lineage it should be better to show the expression of cytokeratins K5, K14 and K19, which also have been used to study lacrimal gland cells (Bannier-Helaouët et al., 2021, Cell Stem Cell 28:1221-1232). Although authors show the expression levels of cytokeratin K5 in their results (Fig 3), it would be convenient that they also show immunostaining of cells with anti-cytokeratins instead of F-actin.

4)    Since the lacrimal gland comprises different cell types, such as acinar, myoepithelial, ductal cells and fibroblasts, the LG cell pool used as a control for RT-PCR experiments, should show as demonstrated in Figure 1, a collection of molecular markers which reflect the presence o the different cell types that constitute the tissue source. However, if the obtained cell clones are derived from the epithelial component of the gland, why is still observed a collection of cell markers that are not typical of epithelial cells such as Vimentin and Smooth muscle actin B2? Such observation would imply that authors are not working with purely lacrimal epithelial cell lines. Please explain.

5)    The permeability barrier, and therefore the epithelial character of the cell line clones was demonstrated by assaying FITC-dextran permeability. However the FITC-dextran used by authors has an average molecular weight of 40,000, which is too high to show that confluent cells on the inserts form an impermeable epithelial cell layer. It would be much better the measurement of the transepithelial electrical resistance (TEER) (Brocke et al., 2024, Physiol Rep. 12(3):e15921; Ghiselli et al., 2021, BMC Mol Cell Biol. 22(1):12), or alternatively use some low molecular weight compounds that could be excluded by a non-permeable epithelial barrier such as ruthenium red or another fluorescent tracer such as FITC-biotin, and showing the constitution of tight junctions by immunostaining of claudins, ZO1 and occludin.

Reviewer 2 Report

Comments and Suggestions for Authors

Round 2

Reviewer 1 Report

Comments and Suggestions for Authors

Authors have complied with all observation made previously. Congratulation for this excellent manuscript.